# A novel assessment method for COVID-19 humoral immunity duration using serial measurements in naturally infected and vaccinated subjects

**Jasper de Boer**[ID][1], **Ursula Saade**[2], **Elodie Granjon**[2], **Sophie Trouillet-Assant**[3,4], **Carla Saade**[3], **Hans Pottel**[1‡], **Maan Zrein**[ID][2‡*], **Covid ser study group**[3¶]

1 Department of Public Health and Primary Care, KU Leuven Campus Kulak, Kortrijk, Belgium, 2 R&D Department, InfYnity Biomarkers, Lyon, France, 3 Virology Laboratory, Institute of Infectious Agents, Laboratory associated with the National Reference Centre for Respiratory Viruses, Civils Hospices of Lyon, Lyon, France, 4 International Center of Research in Infectiology, Institut National de la Santé et de la Recherche Médicale U1111, Centre National de la Recherche Scientifique UMR5308, Ecole Normale Supérieure de Lyon, Claude Bernard Lyon 1 University, Lyon, France

‡ HP and MZ share senior authorship on this work.
¶ Membership of the Covid ser study group is provided in the Acknowledgments.
* mzrein@infynity-biomarkers.com

**Data Availability Statement:** All relevant data are within the paper and its Supporting Information files.

## Abstract

### Background

Collecting information on sustainability of immune responses after infection or vaccination is crucial to inform medical decision-making and vaccination strategies. Data on how long-lasting antibodies against SARS-COV-2 could provide a humoral and protective immunity and prevent reinfection with SARS-CoV-2 or its variants is particularly valuable. This study presents a novel method to quantitatively measure and monitor the diversity of SARS-CoV-2 specific antibody profiles over time.

### Methods

Serum samples from two groups were used in this study: Samples from 20 naturally infected subjects (followed for up to 1 year) and samples from 83 subjects vaccinated with one or two doses of the Pfizer BioNtech vaccine (BNT162b2/BNT162b2) (followed for up to 6 months). The Multi-SARS-CoV-2 assay, a multiparameter serology test developed for the serological confirmation of past-infections, was used to determine the reactivity of six different SARS-CoV-2 antigens. For each subject sample, 3 dilutions (1/50, 1/400 and 1/3200) were defined as an optimal set over the six antigens and their respective linear ranges. This allowed accurate quantification of the corresponding six antibodies. Nonlinear mixed-effects modelling was applied to convert intensity readings from 3 determined dilutions to a single quantification value for each antibody.

### Results

Median half-life for the 20 naturally infected *vs* 74 vaccinated subjects (two doses) was 120 *vs* 50 days for RBD, 127 *vs* 53 days for S1 and 187 *vs* 86 days for S2 antibodies respectively.

**Funding:** The author(s) received no specific funding for this work.

**Competing interests:** Ursula Saade, Elodie Granjon and Maan Zrein are employed by InfYnity Biomarkers. All other co-authors declare no competing interests.

## Conclusion

The newly proposed method, based on a series of a limited number of dilutions, can convert a conventional qualitative assay into a quantitative assay. This conversion helps define the sustainability of specific immune responses against each relevant viral antigen and can help in defining the protection characteristics after an infection or a vaccination.

## Introduction

The severe acute respiratory syndrome coronavirus 2 (SARS-CoV-2) was first described in December 2019 and has rapidly spread. It continues to cause a major health crisis around the world since then [1]. The virus causes the respiratory illness COVID-19, and the outbreak was declared a pandemic on 11 March 2020 by the World Health Organization (WHO) [2]. The COVID-19 pandemic represents the greatest medical and socio-economic challenge of our time [3].

Collecting information on sustainability of immune response after infection is crucial for medical decision-making and vaccination strategies. It is particularly valuable to determine for how long antibodies against SARS-COV-2 could provide humoral immunity and prevent reinfection with SARS-COV-2 or its variants. With the implementation of large-scale vaccination programs, it is critical to determine the duration of vaccine-induced immunity and to anticipate possible modifications of the vaccination strategies. There is currently no consistent depiction of the humoral immune response after natural SARS-COV-2 infection.

The duration of immunity is a key metric of protection following natural infection or vaccination. In addition to cellular immunity, virus-specific antibodies represent an important component of protection. Large efforts are being made globally to set objective criteria to define the degree and duration of such protective immunity at the individual level [4–7]. So far, studies suggest that antibody levels may decrease rapidly in infected individuals. This rate of decrease depends on the severity of the infection (asymptomatic, mildly symptomatic, or severely symptomatic requiring hospitalization and referral to the ICU) [8–10]. Long et al. described a decline in antibody titers in the convalescent phase of the disease, suggesting that antibodies to SARS-CoV-2 may fade away rapidly [8]. Furthermore, Prévost et al. evaluated 98 infected patients and found that while most individuals developed neutralizing antibodies within the first two weeks of infection, the level of neutralizing activity decreased significantly over time [11]. These studies highlight the importance of characterizing the kinetics of antibody levels.

In this study we first present the Multi-SARS-CoV-2 assay (*InfYnity Biomarkers*, Lyon, France). When performed on a minimal dilution sequence, the assay provides a novel method to quantify and monitor SARS-CoV-2 specific antibody profiles over time. The assay can detect the presence of 6 different SARS-CoV-2 specific antibodies in a single well of a 96-well microplate thus reducing inter-assay variability. The method provides a quantitative serological profile, in contrast to conventional immunoassay methods which provide single antibody qualitative results. Second, to demonstrate the effectiveness of the assay, we present an investigation on the difference in antibody waning kinetics in naturally infected versus vaccinated subjects. The presented assay offers a simple and universal method that can be implemented in a wide majority of moderate-resource laboratories.

## Methods

### Study design, population, and origin of the samples

Serum samples were collected from two groups: naturally infected and vaccinated subjects (Fig 1). For the first group 101 serum samples were available, originating from 20 naturally infected

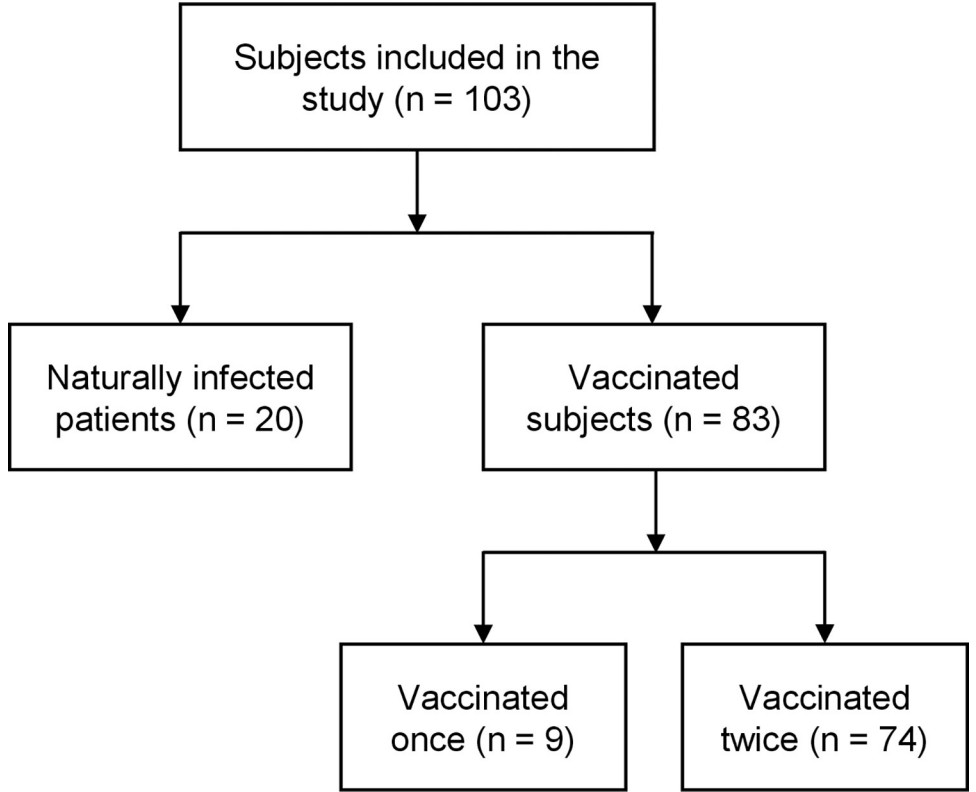

**Fig 1. Subject groups.** The flowchart indicates the grouping of the subjects in the study.

patients, out of which 17 were followed up over time, up to 360 days (median = 268). For each patient 2 to 9 longitudinal samples were available (median = 5). The collection time of the first sample after onset of symptoms was between 14 and 41 days (median = 27.5). All patients were symptomatic, mainly with fever, coughing, having shortness of breath, muscle pain and general weakness. Most patients were confined at home with ambulatory treatment, only a few were hospitalized.

Serum samples from these 20 patients were collected for up to 360 days and tested with the Multi-SARS-CoV-2 immunoassay to demonstrate the monitoring capabilities of the new assay and to investigate the antibody dynamics of naturally infected patients. In addition, serological investigation was performed on the samples using commercial assays: SCoV-2 *Detect*™ IgG ELISA kit for IgG detection from *InBios*, Elecsys® Anti-SARS-CoV-2 test for antibodies detection against the nucleocapsid (N) antigen (including IgG) from *Roche* and LIAISON® SARS-CoV-2 S1/S2 IgG test from *Diasorin*.

Samples from the vaccinated group come from a prospective longitudinal cohort study conducted by the laboratory associated with the National Reference Center for Respiratory Viruses (University Hospital of Lyon, France). Eighty-three health care workers vaccinated with 1 or 2 doses of the Pfizer–BioNTech BNT162b2 vaccine were included in this study. Of these 83 subjects, 74 self-reported no prior COVID infection and were vaccinated twice. The 9 other subjects self-reported prior COVID infection and were only vaccinated once. Blood samples were collected at T0: before the first dose, T1: 4 weeks after the first dose, immediately preceding the second dose, T2: 4 weeks, and T3: 6 months after the full vaccination. The pre-vaccination blood samples were only used to document a previous SARS-CoV-2 infection.

**Table 1. Patient characteristics.**

| Characteristic | Naturally infected group | Vaccinated group |
|---|---|---|
| Sex (Male/Female) | 60% / 40% (12 / 8) | 81% / 19% (67 / 16) |
| Age (Years) | 43 ±[a] 10 (range: 23–60) | 47 ± 11 (range: 22–76) |
| Median # samples | 5 (IQR = 2; range: 2–9) | 4 |
| Follow-up time (days) | 268 (IQR = 89; range: 127–360) | 180 |

[a] ± indicates the standard deviation.

Subjects who have been previously infected with SARS-CoV-2 (convalescent group, 11%) received only one injection, therefore the second sample was omitted (no data at time-point T1). One subject was infected with SARS-CoV-2 between the 2 doses and therefore got a single dose of the vaccine.

Serological investigation was performed on the 323 vaccinated group samples using the Multi-SARS-CoV-2 immuno-assay as well as commercial assay *VIDAS®* SARS-COV-2 IgG II (9COG) for the detection of IgG specific for the SARS-CoV-2 receptor-binding domain (RBD) from *bioMérieux*. Samples at timepoints T0, T1 and T2 were also tested on LIAISON® SARS-CoV-2 TrimericS IgG for the detection of IgG specific to the Spike protein from *DiaSorin*, IgG SARS-CoV-2 (sCOVG) that detects IgG against SARS-CoV-2 from *Siemens* and WANTAI SARS-CoV-2 Ab ELISA for the detection of total antibodies against SARS-CoV-2 from *Wantai*.

Patient characteristics of both subject groups are shown in Table 1.

For each naturally infected patient, 3 dilutions (1/50, 1/400 and 1/3200) at each time-point were assayed. A fourth dilution was available for the vaccinated subjects (1/100), which was used for a qualitative version of the Multi-SARS-CoV-2 assay. In this article, however, we only report the quantitative version of the Multi-SARS-CoV-2 assay.

For the vaccinated subjects, a written informed consent was obtained from all subjects. Ethics approval was obtained from the national review board for biomedical research in April 2020 (Comité de Protection des Personnes Sud Méditerranée I, Marseille, France; ID RCB 2020-A00932-37), and the study was registered on ClinicalTrials.gov (NCT04341142).

For the naturally infected covid patients, immune serum samples collected in 2020 and 2021 were acquired from ABO Pharmaceuticals (ABO Pharmaceuticals, SanDiego, CA, USA) a duly authorized blood banking organization.

None of the subjects were involved in the design or implementation of this study.

## Multi-SARS-CoV-2 assay

The Multi-SARS-CoV-2 immunoassay was used to monitor detailed serological profiles. The multiplex technology allows for the combination of multiple parameters in a single well of a 96-well microplate. The printing process is based on non-contact piezo electric impulsion of a defined volume of an antigenic solution. The bioprinting was performed to print six different SARS-CoV-2 specific antigens: 1) MP1, specific epitope for SARS-CoV-2 membrane antigen, enhanced according to disease severity; 2) NP1, full-length nucleocapsid recombinant protein antigen; 3) NP2, specific epitope of the nucleocapsid protein, enhanced according to disease severity; 4) RBD, recombinant Receptor Binding Domain of the spike protein; 5) S1, recombinant spike protein S1; 6) S2, recombinant spike protein S2.

Antigens were printed at the bottom of each well at precise X-Y coordinates, under controlled humidity and temperature conditions. Each antigen spot is printed in duplicate to improve fault tolerance. Positive control spots were printed in quadruplicate to define a precise

spatial orientation pattern and validate the correct sequential distribution of all biological and chemical materials as well as operating performance (human serum samples, enzyme conjugate, and substrate).

Prior to this study, assay validation was performed in an independent public health laboratory (SCIENSANO, Belgium) and included an extensive testing of well-characterized positive and negative samples to ensure that all performance attributes were in line with in-vitro diagnostic and quality requirements. In brief, each antigen was analyzed for its intrinsic specificity and sensitivity using a set of seropositive (n = 540; all >14 days post-PCR determination) and seronegative samples (n = 270; including pre-pandemic samples). Optimal concentration for each antigen was determined by Receiver Operating Characteristic curves (ROC).

For performance evaluation an NIBSC calibrator was used (ref. 20/136) to define the lower limit of detection for each antibody specificity.

Each plate was read and analyzed using a colorimetric image analyzer. The software calculates the median pixel intensity for each spot with the background noise subtracted. To establish the net intensity for each antigen, the mean value of duplicated spots was calculated. Mean spot intensities for each antigen were normalized by dividing the value by the average positive control intensity, which showed the maximum attainable intensity in the assay. This was done to reduce assay variability and to have a common pixel intensity range of [0, 100] for all tested antigens.

## Dilution sequence

Biomarker reactivity curves that express the relationship between log antibody concentration and intensity typically show a sigmoidal shape that levels off at both ends of the curve. This creates two plateaus one at the low end and one at the high end of the pixel intensity ranges. Consequently, at high and low intensities, a small change in intensity leads to a considerable change in antibody concentration reading. Since minor fluctuations may always be expected due to test variability, the nonlinear plateau-range is ill-suited for an accurate quantification. This is not the case for the linear range that connects the two plateaus and is therefore better suited for quantification.

The Multi-SARS-CoV-2 assay was developed for the serological confirmation of past infections. The assay was therefore optimized to produce strong signals to sera of infected subjects and to show no or non-significant signals to sera from uninfected subjects. Readings in the plateau-ranges are therefore expected.

To ensure a reading of each antigen in the linear range, samples were serially diluted. The dilution factors were chosen in such a way that overlap of linear ranges for all antigens was assured, observed maximal concentrations could be quantified, and the number of dilutions was minimized (to reduce operational cost). This process is explained in more detail in S1 File. Finally, we selected the set of 3 dilutions (1/50, 1/400 and 1/3200) as an optimal set over the 6 antigens. A fourth dilution was available for the vaccinated subjects (1/100).

## Statistical methods

A nonlinear mixed-effects model was used to convert intensity readings from the selected dilutions to a single quantification per antibody. The R 'LME4' package was used for model fitting [12]. The model is based on the standard 2-parameter logistic curve (sigmoidal standard curve with fixed top = 100 and bottom = 0) and is described in the following Eq 1.

$$Intensity_{i,j,k} = \frac{100}{1 + 2^{(log_2(DF50_{i,j}) - log_2(dilution_k)) \cdot hillslope_j}} \tag{1}$$

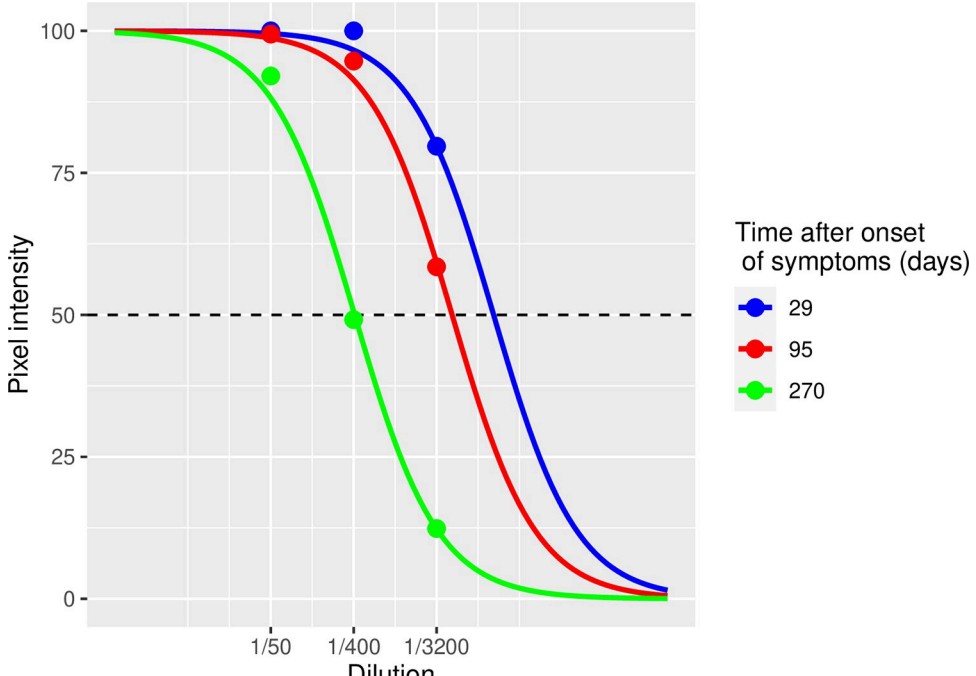

**Fig 2. Sigmoidal curve fitting example.** Sigmoidal curve fitting of 3 dilutions (1/50, 1/400, 1/3200) for 3 timepoints of the naturally infected subject #6 for the anti-RBD antibodies.

In the above equation, $Intensity_{i,j,k}$ corresponds to the intensity of the i[th] sample, j[th] antibody and k[th] dilution. Intensity values were available for I = 424 samples (coming from 103 naturally infected or vaccinated subjects at different time-points), J = 6 antibodies and K = 3 or 4 dilutions. The Dilution Factor 50 (DF50) value corresponds to the estimated dilution at which an intensity value of 50% is observed. This value summarizes the 3 (or 4) antibody intensities and represents quantitative information of antibody concentration. Although the unit of DF50 is arbitrary, its relationship with concentration is linear. The model fit results into 6 (antibody) DF50 values for each individual patients' time-point. Note that the location of the sigmoid curve and thus the DF50 value is obtained from the fitted curve for each sample (thus for each subject at each time-point), based on the measured ODs of the 3 tested dilutions. The hillslopes (per marker) are, however, fitted on the combination of all 424 samples.

As an example, Fig 2 shows the RBD fits for 3 time-points of one naturally infected subject (subject #6). At dilution 1/50 the 29 days, 95 days and 270 days serum samples show high and saturated pixel intensities (or biomarker reactivities) and are thus unquantifiable without the dilution sequence. The fitted sigmoid curves on the three dilutions shift position over time. The position of the curve is described by the DF50-value, that is, the estimated dilution factor corresponding to a pixel intensity of 50. In Fig 2, the DF50s are located at the points where the dashed line at OD = 50 intersects the curves. In the above example, the complete curve clearly shifts over time from right to left, corresponding to a decreasing DF50. This example clearly illustrates the need to use the dilution approach for quantitative measurements.

It is also possible to convert the 2-parameter sigmoidal function to a linear equation, allowing to determine hillslope and DF50 for each individual sample (see S2 File). The plot in this supplement shows an example of the DF50 decay over time for naturally infected patient #1, as calculated with the alternative method.

## Results

In the naturally infected group, antibody concentration generally declines over time. Fig 3 shows the evolution of each (log-transformed) antibody concentration for the 20 patients and 6 markers of the naturally infected group. The general time-evolution trend displays a log-linear decline of concentration, which supports constant half-life for at least the duration of the study period (Fig 3). The graphs also show that decline rates differ between patients and antibodies. A linear regression model was used to fit the log-linear decline per patient and per marker to model the individual dynamics (S1 Table). Table 2 summarizes the decline dynamics for each antibody for the 20 naturally infected subjects. The DF50-values for the first sample of each subject and the calculated half-lives are listed in S1 Table.

In the vaccinated group, antibody concentrations also generally decline over time. The median DF50-values at the different time-points (T0, T1, T2 and T3) for the vaccinated individuals are presented in Table 3. However, among the 74 subjects who were vaccinated twice (Group 1, Table 3), we identified 6 subjects who showed antibody reactivity at baseline (T0). The characteristic times $T_{50}$ and $T_{90}$, obtained from the DF50-values at the two time-points T2 and T3 are summarized in Table 4. These characteristic times are estimated from 2 points only, assuming an exponential decay. As this might be prone to error, we also calculated the relative loss in DF50-reactivity between T2 and T3 (assuming a linear decay between these two time-points). Using RBD as the reference, there is a loss of about 90% in antibodies at time T3, which is 6 months after the second vaccine shot.

When comparing the naturally infected and the vaccinated groups, we found that antibody concentration declines more slowly in the naturally infected group than in the vaccinated group (Tables 2 and 4). In addition, we compared median half-lives for the 20 naturally infected *vs* 74 vaccinated subjects (two doses) using a Wilcoxon rank sum test. Median half-lives were respectively 120 *vs* 50 days for RBD ($p < 0.001$), 127 *vs* 53 days for S1 ($p < 0.001$), and 187 *vs* 86 days for S2 ($p < 0.001$) antibodies.

The raw measurement data of both the naturally infected and vaccinated subjects are available in S3 File. This supplement also contains the results of several commercial assays that were used for clinical screening purposes.

## Discussion

Quantitative measurement of humoral immune response provides an easy and robust surrogate marker of protection. If a natural infection or vaccination induces a sustained and protective immunity, or at least, a long-lasting protection, it may enable the establishment of herd immunity. The presence of antibodies is a key indicator of protective humoral immunity. Anti-RBD and anti-S-protein titers may be particularly important because they correlate with neutralizing activity and are typically associated with early virus control [13–16]. Therefore, the accurate determination of the duration of antibody presence is essential.

Waning antibodies operate in a stealth mode and may not be captured by conventional methods that collectively measure a global and consolidated immune response. Such waning can more easily be detected when monitoring each antibody individually. The novel multiparametric method presented in this study can serve as a tool to measure how individual antibodies wane over time. Serum samples from naturally infected and vaccinated subjects were investigated to examine the effectiveness of our method. We were able to determine the median antibody half-lives for each group (Tables 2 and 4).

Both our results (summarized in Table 2) and several other studies show that the humoral response to SARS-CoV-2 declines over time after natural infection. Zhang et al. reported a significant reduction of humoral response to SARS-CoV-2 within 4 months after diagnosis [17].

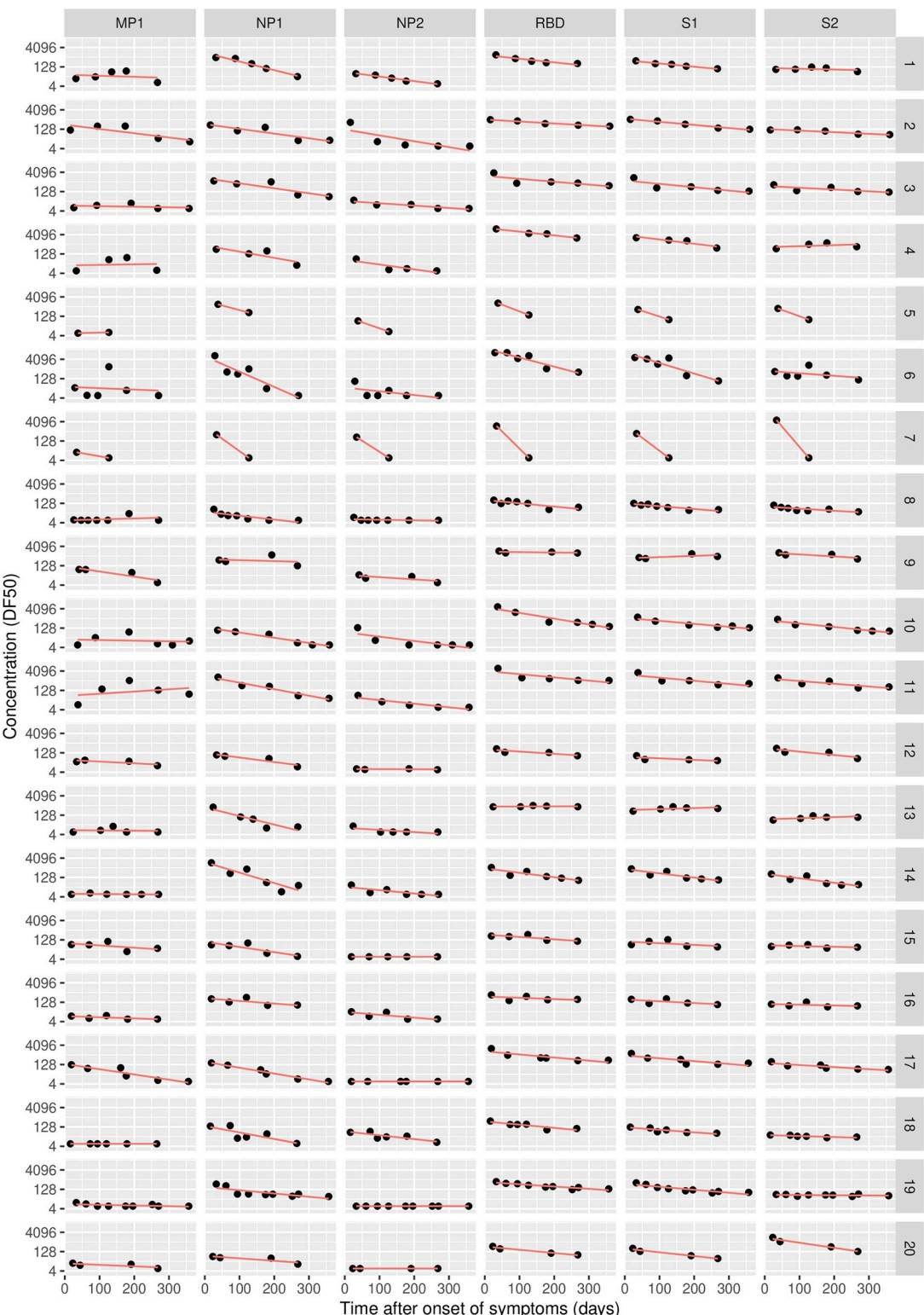

**Fig 3. Antibody concentrations over time for 20 naturally infected subjects.**

**Table 2. Summary of the decline dynamics for 20 naturally infected subjects.**

| Antibody | median DF50 (IQR) of the first sample | median half-life $T_{50}$ in days (IQR) | $T_{90}$ in days (IQR) |
|---|---|---|---|
| MP1 | 11 (6–19) | NE | NE |
| NP1 | 309 (149–536) | 66 (44–79) | 219 (147–264) |
| NP2 | 31 (10–52) | NE | NE |
| RBD | 935 (474–2240) | 120 (87–164) | 398 (289–545) |
| S1 | 461 (230–897) | 127 (91–153) | 420 (302–510) |
| S2 | 262 (93–518) | 187 (112–206) | 620 (374–684) |

Summary of the decline dynamics for each antibody for the 20 naturally infected subjects. DF50 < 50 of the first sample corresponds to no or little reactivity. NE = Not Estimable because of low reactivity.

Studies of Dan et al. and Wu et al. have provided evidence that the circulating immune memory to SARS-CoV-2 appears to reduce over time, but endures for more than 5 months in patients with a previous infection [18, 19]. The observed antibody kinematics for naturally infected subjects in our study correspond well with those reported in literature. Indeed, despite the heterogeneity of immune responses, our limited data indicates sustained (although declining over time) humoral immunity in recovered patients who had symptomatic COVID-19.

More specifically, our estimated RBD half-life can be compared with values previously published in the literature. Dan et al. report that immunoglobulins (Ig)G targeting the SARS-CoV-2 spike protein were found to be relatively stable over time with a half-life of 140 days (95% CI: 20–240 days). Subjects in this study showed mostly mild symptoms [18]. Another study by Kannenberg et al. found an RBD half-life of 158 days (95% CI: 141–181 days) for patients after severe acute respiratory syndrome [20]. Both values are similar to the median RBD half-life found in our study (120 days, IQR: 87–164 days).

It should be noted that the 20 naturally infected patients in our study showed mild to moderate symptoms. Patients who experience high viral replication during COVID-19 infections, typically demonstrate more severe clinical outcomes and show high levels of humoral immunity [21]. However, in this case, our RBD half-lives appear similar to the ones reported by Kannenberg et al. for severe acute respiratory syndrome [20].

**Table 3. Summary of the dynamics for each antibody for the 83 vaccinated subjects.**

| Group | Antibody | T0 | T1 | T2 | T3 |
|---|---|---|---|---|---|
| 1 (n = 74) | RBD | ≤6 (≤6–8) | 261 (122–551) | 3123 (2225–7076) | 460 (277–735) |
| | S1 | ≤6 (≤6 – ≤6) | 104 (53–205) | 1530 (873–2727) | 245 (140–453) |
| | S2 | ≤6 (≤6–12) | 32 (18–56) | 109 (51–193) | 29 (20–53) |
| 2 (n = 9) | RBD | 148 (42–257) | - | 6448 (5650–10316) | 1473 (1264–716) |
| | S1 | 60 (20–66) | - | 4773 (3384–5488) | 833 (716–865) |
| | S2 | 45 (27–62) | - | 1501 (936–1626) | 126 (74–164) |
| 3 (n = 6) | RBD | 38 (22–67) | 3121 (144–5936) | 4018 (3339–8527) | 1211 (527–2910) |
| | S1 | 15 (8–33) | 1802 (56–4317) | 2278 (1466–7109) | 708 (375–1890) |
| | S2 | 21 (12–42) | 388 [39–1283] | 327 (118–1028) | 108 (57–208) |

Median DF50-values (IQR) for RBD, S1 and S2 are presented at the different time-points T0 (baseline), T1 (3 weeks after first vaccine), T2 (1 month after second vaccine) and T3 (6 months after second vaccine) for group 1 (n = 74) vaccinated subjects who had no prior COVID infection, for group 2 (n = 9) vaccinated subjects who had a prior COVID infection and for group 3 (n = 6) vaccinated subjects who reported no prior COVID infection, but who showed antibodies reactivity at baseline (T0). The other antibodies (MP1, NP1 and NP2) were not reported as they were not triggered by the vaccines.

**Table 4. Characteristic times for the waning antibodies.**

| Group | Antibody | $T_{50}$ in days (IQR) | $T_{90}$ in days (IQR) | Estimated % relative loss between T2 and T3 |
|---|---|---|---|---|
| 1 (n = 74) | RBD | 50 (42–58) | 168 (140–193) | 87% (82–91) |
| | S1 | 53 (45–68) | 177 (149–227) | 86% (77–90) |
| | S2 | 86 (59–123) | 287 (197–309) | 69% (56–81) |
| 2 (n = 9) | RBD | 55 (48–61) | 182 (160–203) | 84% (76–86) |
| | S1 | 49 (46–65) | 164 (153–214) | 84% (71–89) |
| | S2 | 37 (37–51) | 122 (121–168) | 92% (71–94) |
| 3 (n = 6) | RBD | 84 (62–102) | 278 (205–340) | 71% (64–82%) |
| | S1 | 82 (78–91) | 272 (258–302) | 72% (68–74%) |
| | S2 | 122 (94–155) | 404 (312–514) | 58% (49–67%) |

The half-life $T_{50}$ ($T_{90}$) equals the number of days at which there is 50% (10%) left from the estimated DF50 at time zero assuming an exponential decay. The estimated % relative loss is calculated as [DF50(T2)—DF50(T3)] / DF50(T2). We compared specific antibodies between groups 1 and 2 using a Wilcoxon rank sum test: RBD, p = 0.330; S1, p = 0.855: S2, p = 0.001.

In the vaccinated group, we found that our results (summarized in Table 4) correspond well with published studies. Achiron et al. report an estimated half-life of 45 days for the S1 antibody [22]. Doria-Rose estimated a RBD half-life of 52 days (95% CI: 46–58 days) from a cohort vaccinated twice with the mRNA1273 (Spikevax) vaccine [23]. Both these half-life values are comparably close to our estimate of 53 days.

Our results show that the initial humoral response is stronger in the vaccinated group, but declines much faster, than in the naturally infected group (Tables 2 and 4). Half-lives were significantly shorter for every antibody. For example, the RBD median half-life was 50 days in the group that received 2 vaccinations versus 120 days for the naturally infected group. This faster antibody titer decline in vaccinated groups versus naturally infected groups, corresponds with results previously published in literature. For example, larger declines in antibody titers have previously been observed in subjects vaccinated with the Pfizer BioNtech vaccine versus naturally infected subjects [24].

Our results indicate that the Multi-SARS-CoV-2 assay can distinguish the humoral response to natural infections from the response to vaccination. In vaccinated subjects, antibodies targeting MP1, NP1 and NP2 were typically absent, while this is not the case for the naturally infected group. This is in line with the nature of the BNT162b2/Comirnaty vaccine because it is solely based on the spike protein antigens [25].

A potential future area of study is to investigate the role of the individual antibodies measured in the Multi-SARS-CoV-2 assay. While anti-RBD and anti-S antibodies may be related to neutralization activity, the precise role of the membrane protein antigen (MP1) and the nucleocapsid recombinant protein antigens (NP1 and NP2) still need to be determined. Our analyses on the naturally infected group showed that the estimated half-life of the nucleocapsid recombinant protein antigen NP1 was only about half of the estimated half-life of RBD, S1 and S2. In vaccinated subjects who had a prior COVID infection, baseline values of NP1 and NP2 were similar (that is, quite low) to those of the baseline values of subjects who did not declare a prior COVID infection. This suggests that these antibodies disappear much faster after infection than the other antibodies.

## Conclusions

The correspondence in antibody kinematics of our results and the data found in the literature, both for naturally infected and vaccinated subjects, supports the validity of our newly

presented method. The method, based on a series of a limited number of dilutions, can convert a conventional assay from qualitative testing into a quantitative assay. This enables the quantification of the time decay of antibody response as well as the calculation of the half-life for these antibodies. This new procedure provides information on the sustainability of immune response, helping us to understand and estimate the duration of humoral immunity, after infection or vaccination.

## Supporting information

**S1 Table. Individual results for the 20 naturally infected subjects.** Dilution Factors corresponding to 50% reactivity for the first sample is presented for the 6 different antibodies. First and Last sample* indicates the time lapse in days since first symptoms of infection.
N = number of samples for each subject. "≤ 6" means that the DF50-value was estimated below limit of quantification. Half-lives are obtained from the linear regression model of ln (DF50) against time in case the model returned a negative slope. Half-life greater than last sample time or when the model returned zero or positive slopes were annotated as > last sample day. Half-lives were indicated as Non-Estimable (NE) when slopes could not be estimated due to low first sample DF50 values (≤50).
(DOCX)

**S1 File. Deriving an optimal dilution sequence.**
(DOCX)

**S2 File. Alternative fitting method.**
(DOCX)

**S3 File. The dataset.** Excel database containing pixel intensities for 424 samples, 6 markers and several dilutions. All the results obtained from commercial assays (VIDAS, DIASORIN, SIEMENS AND WANTAI) correspond to a working dilution as recommended by the corresponding manufacturer. The dilution sequence method has been performed only for the Multi-SARS-CoV2 kits.
(XLSX)

## Acknowledgments

The Covid-Ser study group composed of Kahina Saker, Christelle Compagnon, Bouchra Mokdad, Virginie Pitiot, Cecile Barnel, Vanessa Escuret, Florence Morfin, Mary-Anne Trabaud, Laurence Josset, Dulce Alfaiate, Jean-Baptiste Fassier, Alexandre Gaymard, Grégory Destras, Nicolas Guibert, Hélène Lozano and Amélie Massardier Pilonchery Collected the samples and performed experiments.

Group leader and contact: Sophie Trouillet-Assant, email: sophie.assant@chu-lyon.fr

## Author Contributions

**Conceptualization:** Hans Pottel, Maan Zrein.

**Data curation:** Ursula Saade, Elodie Granjon.

**Formal analysis:** Jasper de Boer, Hans Pottel.

**Investigation:** Ursula Saade, Elodie Granjon.

**Methodology:** Jasper de Boer, Ursula Saade, Elodie Granjon, Hans Pottel, Maan Zrein.

**Project administration:** Hans Pottel, Maan Zrein.

**Resources:** Sophie Trouillet-Assant, Carla Saade.

**Software:** Jasper de Boer, Hans Pottel.

**Supervision:** Hans Pottel, Maan Zrein.

**Validation:** Hans Pottel, Maan Zrein.

**Visualization:** Jasper de Boer.

**Writing – original draft:** Jasper de Boer, Ursula Saade, Hans Pottel, Maan Zrein.

**Writing – review & editing:** Jasper de Boer, Ursula Saade, Elodie Granjon, Sophie Trouillet-Assant, Carla Saade, Hans Pottel, Maan Zrein.

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
