## [Decision Letter · Decision Letter 0]

27 Apr 2022

PONE-D-22-04042A novel assessment method for COVID-19 humoral immunity duration using serial measurements in naturally infected and vaccinated subjectsPLOS ONE

Dear Dr. Boer, 

Thank you for submitting your manuscript to PLOS ONE. After careful consideration, we feel that it has merit but does not fully meet PLOS ONE’s publication criteria as it currently stands. Therefore, we invite you to submit a revised version of the manuscript that addresses the points raised during the review process. Both reviewers believes that the manuscript is interesting. However, both one have concerns about the writing style particularly the discussion is week. I also support reviewer 2 comments that you should compare your assay results to other commercially available kits to evaluate the performance of your assay.  Please submit your revised manuscript by Jun 11 2022 11:59PM. If you will need more time than this to complete your revisions, please reply to this message or contact the journal office at plosone@plos.org. Please include the following items when submitting your revised manuscript:A rebuttal letter that responds to each point raised by the academic editor and reviewer(s). You should upload this letter as a separate file labeled 'Response to Reviewers'.A marked-up copy of your manuscript that highlights changes made to the original version. You should upload this as a separate file labeled 'Revised Manuscript with Track Changes'.An unmarked version of your revised paper without tracked changes. You should upload this as a separate file labeled 'Manuscript'.

We look forward to receiving your revised manuscript.

Kind regards,

Gheyath K. Nasrallah

Academic Editor

PLOS ONE

Journal Requirements:

3. One of the noted authors is a group or consortium “Covid ser study group”. In addition to naming the author group, please list the individual authors and affiliations within this group in the acknowledgments section of your manuscript. Please also indicate clearly a lead author for this group along with a contact email address.

Reviewers' comments:

Reviewer's Responses to Questions

**Comments to the Author**

1. Is the manuscript technically sound, and do the data support the conclusions?

Reviewer #1: Partly

Reviewer #2: Partly

2. Has the statistical analysis been performed appropriately and rigorously? 

Reviewer #1: I Don't Know

Reviewer #2: Yes

3. Have the authors made all data underlying the findings in their manuscript fully available?

Reviewer #1: Yes

Reviewer #2: No

4. Is the manuscript presented in an intelligible fashion and written in standard English?

Reviewer #1: No

Reviewer #2: No

5. Review Comments to the Author

Reviewer #1: The manuscript submitted by Jasper de Boer and colleagues, titled “A novel assessment method for COVID-19 humoral immunity duration using serial measurements in naturally infected and vaccinated subjects” presents a new assessment method for COVID-19 humoral immunity duration which can convert a conventional qualitative assay into a quantitative assay using serial measurements. Although this is a worthwhile study, the manuscript needs a major revision and contains several weak points that authors should address before the manuscript is published. While I understand that English is likely not the authors' first language, the manuscript would greatly benefit from grammatical and verbiage revisions from one more skilled in English. I highlighted a few of the errors below.

Minor Comments:

1. Pages 2, please rephrase the aim clearly in the Abstract, “This study aimed to propose a novel method to ..”

2. Page 2, the word “respectively” is placed at the middle of the sentence, it must be at the end.

3. Page 3, an APA in-text citation should be placed before the final punctuation mark in a sentence, preferably, please move citation #1 to the end of the sentence.

4. Page 3, please delete the sentence: “As there is no antiviral drug to treat the disease at this moment”

5. Page 3, please rephrase “ now that massive …” to “With the implementation of large-scale vaccination programs, it is critical to determine the duration of vaccine-induced immunity …"

6. Page 3, please rephrase “At this moment, a consistent depiction of …” to “There is currently no consistent depiction of …. ”

7. Page 3, second paragraph, there is a left parenthesis but no corresponding right parenthesis, please fix the error.

8. References should be presented according to journal guidelines. Referencing is extremely poor across the manuscript. For example, citations 3, 4, and 5 are not cited in the correct format, they should be merged, please revise all in-text citations across the manuscript.

9. Page 3, the sentence “The duration of immunity is, of course, a very important ..” looks informal, please rephrase.

10. Page 4, please change the word “collection”, I suggest rephrasing the sentence to “serum samples were collected from two groups: vaccinated and naturally infected subjects”

11. Page 4, please add a comma before “out of which”

12. Please rephrase “Patients were all symptomatic” to “All of the patients were symptomatic”.

13. Please change “collection” to “group”, throughout the manuscript.

14. Page 5, in the sentence “before the second injection of vaccine corresponding to 4 weeks after the first dose for participants”, please remove “for participants”.

15. Please be consistent when referring to study subjects, either use “subjects” or “participants”.

16. I suggest deleting unnecessary subtitles under the “Methods” section, such as “Ethical Statements” and “Patient and public involvement”

17. Please make sure to follow the journal's instructions for preparing the tables, particularly with regard to aspects such as the numbering style, titles, formats, etc.

18. Page 10, last paragraph, please add a space after the bracket in “(self-)reported no prior”.

19. Page 10, last paragraph, please rephrase “Taking RBD …” to “using RBD …”

20. Please be consistent with spacing around dashes and numbers, for example in Table 2, please add a space after the dash in “620 (374 –684)”, also in Table 3 in “32[18– 56]”, “708 [375– 1890]”, “1211 [527– 2910]” , and “108 [57– 208]”.

21. Page 14, the first paragraph in the discussion section is very long, please split it into 2-3 paragraphs.

22. Page 15, please change “can easily be applied” to “can be easily applied”

23. The conclusion can be improved. The conclusion should help the readers understand why your research is significant to them. It should be a synthesis of the main concepts, where you may propose new areas for future research investigations if applicable, rather than a recap of the main points covered or a re-statement of your research issue. Also, some sentences are not properly written in terms of clarity and language.

24. An APA in-text citation is placed before the final punctuation mark in a sentence, please carefully revise and fix all in-text citations.

25. In Figure 1, please change “days since symptoms” to “Time after onset of symptoms (days)”

Major comments:

1- A major drawback to the manuscript is that the standard of English is significantly sub-par and makes it very difficult to read, obscuring the important message being conveyed. This needs considerable review to make it suitable for publication.

2- The discussion section lacks a detailed discussion of the results, the tables and figures, deserve further explanation in the Discussion section, please cite the tables and figures where appropriate. The discussion should summarize the main findings, compare the findings with other studies, and discuss the implication of the findings and conclusions. The discussion lacks proper referencing and comparison. There are a lot of statements (particularly in pages 16 and 17), which need to be supported by references to increase credibility of the information provided.

3- The discussion extremely lacks coherence and flow of thought. The whole section needs to be divided into shorter paragraphs that flow logically from one idea to the next. The manuscript lacks a clear topic sentence at the beginning of each paragraph that would help improve the flow of the paragraphs. For example, in the last paragraph of page 16, the authors started the paragraph with a very poor and extremely short statement “Our study confirms these last findings.” Also, other paragraphs, such as “Studies of Dan et al. and Wu et al. have provided … with previous SARS- CoV-2 infection” are extremely short and dull. The most important sentence in a paragraph is generally the topic sentence, which should clearly state the subject of the whole paragraph. A paragraph should have the following elements to be as effective as possible: unity, coherence, a topic sentence, and adequate development. All of these aspects overlap. Most importantly, an entire paragraph should be focused on a single topic. It should not finish with another or meander between various concepts.

Reviewer #2: The authors of this manuscript are presenting an immunoassay assessment method to measure multiple SARs-CoV-2 antibodies over a period of time.

Commercial kits that can detect, quantify, and even characterize the neutralizing properties of SARS-CoV-2 antibodies are already available. Some of these kits provide multiplex panels to quantitate multiple SARS-CoV-2 antibodies in a single reaction. Moreover, there are kits designed to detects neutralizing antibodies against different variants of SARS-CoV-2.

So, one of the major concerns in this article is how different is their method compared to other available kits/methods. The authors should have compared their method with other commercially available kits to evaluate the sensitivity of their test. The introduction is generally well written, and the aim is clearly stated.

Methods:

Please mention the collection time (period) of first sample following natural infection.

In page 5, it is not enough to rely on pre-vaccination blood to exclude previous infection. This should be coupled with patient history.

Also, in page 5, it is mentioned that “commercial assays were performed”. Which assays and used for what and where is the results. It is important to expand on this.

In page 6, which S antigens were use for the assay. For which variant. Please specify.

Also in page 6, it mentioned that a validation test was used to validate the immunoassay. Is this a published data. If so, please cite and if not, please provide more details even as supplementary.

The same thing applies for dilutions selected at the end of page 7. Either add a reference number of more details to show how exactly you reached to the conclusion to use these 3 or 4 dilutions.

In page 8, I think it should be j=6 not 7.

Discussion:

Authors are listing the results of their study and others without comparison in the first two pages.

In page 14, please add reference about the link between severity and level of antibodies.

In page 15, it is not reliable to compare NP from vaccinated (with no previous infection) and infected individuals.

Also in the last paragraph of page 15, “estimated half-life of RBD, S1, S2” which group?? And half-life reported in this study should be compared to those reported in other studies/ by other methods.

In page 16 is an example. And why they are talking about nAbs here. This study did not test for neutralizing activity of the antibodies.

6. PLOS authors have the option to publish the peer review history of their article (what does this mean?). If published, this will include your full peer review and any attached files.

Reviewer #1: No

Reviewer #2: **Yes: **Hebah Al Khatib

---

## [Author Response · Author response to Decision Letter 0]

22 Jun 2022

We have responded to the comments in our rebuttal letter.

---

## [Decision Letter · Decision Letter 1]

23 Aug 2022

PONE-D-22-04042R1A novel assessment method for COVID-19 humoral immunity duration using serial measurements in naturally infected and vaccinated subjectsPLOS ONE

Dear Dr. Boer

Thank you for submitting your manuscript to PLOS ONE. After careful consideration, we feel that it has merit but does not fully meet PLOS ONE’s publication criteria as it currently stands. Therefore, we invite you to submit a revised version of the manuscript that addresses the points raised during the review process. We are very close to acceptance. One Reviewer 1 has some comments that you need to address promptly before I endorse publications.  

We look forward to receiving your revised manuscript.

Kind regards,

Gheyath K. Nasrallah

Academic Editor

PLOS ONE

Journal Requirements:

Reviewers' comments:

Reviewer's Responses to Questions

**Comments to the Author**

1. If the authors have adequately addressed your comments raised in a previous round of review and you feel that this manuscript is now acceptable for publication, you may indicate that here to bypass the “Comments to the Author” section, enter your conflict of interest statement in the “Confidential to Editor” section, and submit your "Accept" recommendation.

Reviewer #1: All comments have been addressed

Reviewer #2: All comments have been addressed

2. Is the manuscript technically sound, and do the data support the conclusions?

Reviewer #1: Yes

Reviewer #2: Yes

3. Has the statistical analysis been performed appropriately and rigorously? 

Reviewer #1: Yes

Reviewer #2: I Don't Know

4. Have the authors made all data underlying the findings in their manuscript fully available?

Reviewer #1: Yes

Reviewer #2: Yes

5. Is the manuscript presented in an intelligible fashion and written in standard English?

Reviewer #1: Yes

Reviewer #2: Yes

6. Review Comments to the Author

Reviewer #1: Manuscript Number: PONE-D-22-04042R1

A novel assessment method for COVID-19 humoral immunity duration using serial measurements in naturally infected and vaccinated subjects

PLOS ONE

Dear Editor, dear Authors,

I have carefully read the revised version of the manuscript submitted by Jasper de Boer and colleagues, titled “A novel assessment method for COVID-19 humoral immunity duration using serial measurements in naturally infected and vaccinated subjects”.

This paper reports interesting results regarding a new assessment method for COVID-19 humoral immunity duration which can convert a conventional qualitative assay into a quantitative assay using serial measurements.

All of my previous comments have been addressed by the authors, and I believe the paper is in a good way for publication.

Reviewer #2: The authors have modified the manuscript as requested in the first revision.

Methods Thanks for explaining in detail the characteristics of the groups. I would suggest making a chart for easier tracking.

Result

o Please replace antigen to antibodies in the results

o Table 4, group 3, estimated %relative, please use as percentages as for groups 1 and 2.

o Good that you compared antibody titers using other kits, however, according to S4 table there are no differences in titers among the three dilutions when tested by the commercial kits. Your assay on the other hand is showing a decrease in titer compared to your assay. Is there an explanation for this? Please clarify this in results. Also, please add a sentence or two to describe how different are results of assays (especially ones that target S1/S2) compared to your assay.

Discussion:

o Add “anti” before “RBD and S-protein titers may be particularly important …”

o Please rephrase the following paragraph. It is not clear. “These results correspond well with results previously published in literature. Large differences in antibody titer declines have previously been observed between naturally infected subjects and subjects vaccinated with the Pfizer BioNtech vaccine [24]. Moreover, as discussed above, half-life results from both vaccinated and naturally infected groups generally correspond with the literature.….”

7. PLOS authors have the option to publish the peer review history of their article (what does this mean?). If published, this will include your full peer review and any attached files.

Reviewer #1: **Yes: **Salma Younes

Reviewer #2: No

---

## [Author Response · Author response to Decision Letter 1]

30 Aug 2022

Dear editor and reviewers,

Our response to the raised points are addressed in the rebuttal letter.

---

## [Editor Report · Decision Letter 2]

31 Aug 2022

A novel assessment method for COVID-19 humoral immunity duration using serial measurements in naturally infected and vaccinated subjects

PONE-D-22-04042R2

Dear Dr. Boer,

We’re pleased to inform you that your manuscript has been judged scientifically suitable for publication and will be formally accepted for publication once it meets all outstanding technical requirements.

Kind regards,

Gheyath K. Nasrallah

Academic Editor

PLOS ONE
---

## [Editor Report · Acceptance letter]

20 Sep 2022

PONE-D-22-04042R2 

A novel assessment method for COVID-19 humoral immunity duration using serial measurements in naturally infected and vaccinated subjects 

Dear Dr. Zrein:

I'm pleased to inform you that your manuscript has been deemed suitable for publication in PLOS ONE. Congratulations! Your manuscript is now with our production department. 

Kind regards, 

on behalf of

Dr. Gheyath K. Nasrallah 

Academic Editor

PLOS ONE